# A Community Bundle to Lower School-Aged Obesity Rates in a Small Midwestern City

**Allison Barnes [1], Michelle E. Hudgens [1], Debora Robison [2], Roger Kipp [3], Kathleen Strasser [2] and Robert M. Siegel [1,4,*]**

1   The Heart Institute, Cincinnati Children's Hospital, Cincinnati, OH 45229, USA
2   Center for School-Based Mental Health Programs, Miami University, Oxford, OH 45056, USA
3   Norwood City School District, Norwood, OH 45212, USA
4   College of Medicine, University of Cincinnati, Cincinnati, OH 45212, USA
*   Correspondence: Bob.siegel@cchmc.org; Tel.: +1-513-636-9420; Fax: +1-513-636-2459

**Abstract:** Background: Multi-component interventions in large communities such as Philadelphia can effectively lower childhood obesity rates. It is less clear whether this type of intervention can be successful in smaller communities with more limited resources. Norwood, Ohio is a small Midwestern city with a population of 19,207. In 2010, Ohio passed a school health law requiring Body Mass Index (BMI) screening of students in kindergarten and grades 3, 5 and 9 along with restrictions on competitive foods and vending machine products and a physical education requirement of 30 min per day. In 2014, Norwood implemented a multi-component childhood obesity prevention and treatment bundle of interventions. Our objective was to describe the effects if this bundle on childhood overweight/obesity (OW/OB) rates. We hypothesized that implementation of the bundle would lower the prevalence of OW/OB in Norwood school children. Methods: In 2012, the Healthy Kids Ohio Act was fully implemented in the Norwood City School District (NCSD). In 2014 a comprehensive bundle was implemented that included: 1. A student gardening program; 2. Supplementation of fresh produce to a local food pantry and a family shelter; 3. A farmers market; 4. A health newsletter; 5. Incentives in the school cafeterias to promote healthy food selection; 6. A 100-mile walking club; 7. "Cook for America" (a "cooked from scratch" intervention for school cafeterias); 8. A school-based obesity treatment clinic; Results: The OW/OB rate in the NCSD was 43% at the time of the Bundle implementation in 2014 and 37% in 2016 ($p = 0.029$). Conclusions: A childhood OW/OB prevention bundle can be implemented in a small city and is associated with a favorable change in BMI.

**Keywords:** children; obesity; community; bundle

## 1. Introduction

Obesity continues to be an increasing problem in the US for both adults and children [1,2]. The causes of childhood obesity are complex and risk factors include socioeconomic status, environmental factors, inter-uterine exposures, genetic predisposition, perinatal-antibiotic exposure and stress [3–5]. Unfortunately, interventions that focus on individual behavior changes have had limited impact and this has led to greater interest in wider, community-based interventions that more definitely address environmental factors [6,7]. With this in mind and recognizing that many factors led to the dramatic increase in overweight and obesity (OW/OB) rates, several communities implemented multi-component programs or "Bundles" to reverse this alarming trend [8–10]. In 2010, the Center for Disease Control (CDC) further stimulated these efforts with "Putting Prevention to Work" (PPW) initiative, in which 50 communities nation-wide received funding for obesity and tobacco use prevention [11,12]. Three

community bundles to address obesity in relatively large cities (greater than 75,000 population) Cambridge, Somerville, and Philadelphia are noteworthy in that body mass index (BMI) in school children were significantly lowered [13–16]. A concern for smaller sized communities is that they may have fewer resources to implement a multi-component program. We hypothesized that implementation of a multi-component bundle would lower the prevalence of OW/OB in Norwood school children. With this study, we describe the successful efforts of a small Midwestern city in implementing a multi-component "bundle" to prevent childhood obesity.

Norwood, Ohio is a small Midwestern city of 19,207 which is surrounded by the larger municipality of Cincinnati, Ohio, population 296,943 [17,18]. In the fall of 2013, the Norwood community along with the advisory help of the Center for Better Health and Nutrition (CBHN) of Cincinnati Children's Hospital held a health summit to generate ideas for a multi-component bundle to lower the OW/OB rate in children. A leadership committee of community members, school personnel and advisors from the CHBN met monthly and ultimately planned a bundle of interventions (Table 1). Previously, obesity prevention efforts included an Ohio statewide school health act (passed in 2010, fully implemented 2012) and a countywide initiative funded through the Center for Diseases Control (2010) which are summarized in Table 2. Thus the "Norwood Bundle" was a multi-component intervention in addition to statewide and countywide initiatives. An anonymous family foundation gift of $30,000 helped underwrite the cost of the "Norwood Bundle."

**Table 1.** Summary of the components of the "Norwood Bundle".

| Bundle Element | Number of People Affected |
| --- | --- |
| Center for Better Health and Nutrition Clinic | 18 visits per year |
| Lydia's House/Zion Food Pantry (Fresh Produce) | 327 people per month |
| Woven Oaks | 100 children per year |
| 100 mile Club | 120 students per year |
| Farmer's Market | 400–500 people per year |
| Cook for America | 1200 students per year |
| Newsletter | 4500 people per year |
| Power Plate | 600 students per year |

**Table 2.** Summary of Ohio State School Health Act—"Healthy Choices for Healthy Kids" and the CDC funded "We Thrive" initiative.

| Healthy Choice for Healthy Kids-Goals |
| --- |
| • To offer more nutritious foods and beverages during the regular and extended school day in vending machines and other school-operated venues. |
| • To improve food and beverage nutrition standards for the school lunch program |
| • To implement a school-wide program that offers 30 min of exercise daily at each school |
| • Starting in 2013, physical education teachers hired must be certified |
| • Schools are to develop a core health education curriculum centered on the benefits of physical activity and nutrition |
| • BMI screening is to be done at K, 3rd, 5th and 9th grades with results sent to parents |
| • Schools to submit an annual report to the Ohio Department of Health on students' BMI |

| We Thrive-Goals |
| --- |
| • To support healthier vending machine foods as described in the School Health Act |
| • To increase community gardens |
| • To encourage medical primary care providers to improve preventive obesity care |
| • To support safe routes to schools |
| • To collect data on high school students with the Youth Risk Behavior Survey |
| • To support YMCA (Young Man's Christian Association) membership |

## 2. Methods

The "Norwood Bundle" of initiatives implemented in the Fall of 2014 that are summarized in Table 1 and detailed below:

*The School CBHN Clinic*—(The Center for Better Health and Nutrition [CBHN]) is the pediatric weight management at Cincinnati Children's Hospital and was invited by Norwood City School District (NCSD) to bring their program to the centrally located high school building. NSCD has three elementary schools, a middle school, and a high school. A stage-three structured pediatric weight management clinic was established as described by the American Academy of Pediatrics with a CBHN multi-disciplinary team of nurse, pediatrician, dietitian and exercise physiologist. [19] The clinic was open to any student attending NCSD with a BMI ≥ 85th percentile for age and sex. The clinic was held monthly and up to four students and their guardians were cared for at each clinic session. The students and families were scheduled visits every two months, averaging 18 visits per year.

*Fresh Fruit and Vegetable Supplements to Lydia's House and Zion Food Pantry*—Lydia's House provides transitional housing to women and children in crisis. The Zion Food Pantry is a food pantry located in Norwood and services those in need with supplemental food on four Tuesday mornings a month. Over a two-year period, $10,000 were allotted to Lydia's House and the Zion Food Pantry to supplement fresh fruits and vegetables to their food offerings. The number of people accessing these organizations was about 327 per month.

*Woven Oaks*—A Norwood based organization that fosters gardening skills and knowledge about nature. Its programs include gardening classes called "Norwood Grows" for children ages 11 to 14. The bundle included funding to support this program in which about 100 children participated each year.

*The 100 Mile Club*—Is an afterschool program that promotes physical activity in elementary school children. Participants in the club are asked to log 100 miles walking or running over the course of an academic school year both during school and out school. The bundle funded t-shirts and other incentives to children who participated in the program. About 100 children participated in the club each year of the intervention.

*The Norwood Farmer's Market*—A market of local vendors that meets monthly from June through September. The market features fresh produce, other healthy foods, and crafts. An estimated 400 to 500 people attended during those months during the intervention. The bundle proved funds to help establish the farmer's market.

*Cook for America*—A program to assist schools food services in order to improve the quality and nutrition of cafeteria food. Under the program, at least 60% of the food is "scratch-cooked" on site. This program affected about 1200 students per year of the intervention and was funded by a separate grant from *Interact for Health.*

*Community Newsletter*—One of the issues noted at the community health summit was the lack of a consistent source of health information for the community. With this in mind, a newsletter was started that featured local health events and activities that promoted a healthier lifestyle in the Norwood community. The newsletter also featured stories on health and nutrition. The newsletter was distributed at the Norwood schools and by local merchants. The distribution each quarter was 4500.

*The Power Plate Program (PPP)*—This program was done in the NCSD elementary schools to promote the healthiest choice selection for lunch in the school cafeteria during the intervention. Children had the opportunity to get a sticker or temporary tattoo 2 days a week if they selected the "Power Plate" consisting of an entrée with whole grain, a fruit, a vegetable, and plain milk on those days. A separate analysis of the PPP demonstrated >200% increase in "Power Plate" selection without increasing food waste [20,21].

The potential impact of the "Norwood Bundle'" was objectively evaluated by the measuring the BMI of the children of NSCD and comparing the prevalence of OW/OB to that of Cincinnati Public School (CPS) children determined through their BMI screening. BMI data for students in grades K, 1, 3, 5, and 9 were made available for NCSD and the surrounding school district, CPS, through the *"Healthy Choices for Healthy Kids Act (HCHKA)"*. Height and weight were measured with the children in light clothing and no shoes using a portable scale and stadiometer (SECA, Hanover, New Hampshire). Overweight was defined by the Center for Disease Control and Prevention (CDC) recommendation of the 85th percentile of BMI for age and sex to ≤ the 95th percentile and obesity as ≥ the 95th percentile;

the children's weight status was determined using the Excel spreadsheet offered by the CDC [22]. CPS was used as a comparison group because of its close proximity and because true control was not possible since the "Norwood Bundle" was a community-wide intervention. The demographics and differences of the two school districts are shown in Table 3 [23,24]. Chi-Square analysis (Medcalc ver 18.5, Ostend, Belgium) was used to test the differences of the OW/OB prevalence between the full implementation of the bundle during the 2014–2015 school year and 2016–17 and the demographics of the NSCD and CPS students. The institutional review board of Cincinnati Children's Hospital determined this study to be exempt from review in accordance with applicable regulations and institutional policy.

**Table 3.** Demographics of Norwood City School District and Cincinnati Public Schools.

| School District | Norwood City | | Cincinnati Public | | |
|---|---|---|---|---|---|
| Category | Number | Percent | Number | Percent | *p* Value |
| All Students | 1959 | 100 | 34,816 | 100 | - |
| American Indian or Alaskan Native | - | NC | 38 | 0.1 | 0.14 |
| Asian or Pacific Islander | - | NC | 522 | 1.5 | <0.0001 |
| Black, Non-Hispanic | 270 | 13.8 | 21802 | 62.6 | <0.0001 |
| Hispanic | 238 | 12.2 | 2036 | 5.8 | <0.0001 |
| Multiracial | 109 | 5.6 | 2183 | 6.3 | 0.21 |
| White, Non-Hispanic | 1337 | 68.2 | 8236 | 23.7 | <0.0001 |
| Students with Disabilities | 333 | 17 | 6541 | 18.8 | 0.048 |
| Economic Disadvantage | 1305 | 66.6 | 28,527 | 81.9 | <0.0001 |
| English Learner | 90 | 4.6 | 2087 | 6.0 | 0.011 |

NC = Too low to calculate.

## 3. Results

BMI data of NCSD and CPS are displayed in Figure 1; Figure 2. Between 602 and 646 children were measured in NCSD per year and the larger system of CPS had between 1644 and 9181 measurements per year. Of note, 41% of NCSD children were OW/OB after the Healthy Choices for Healthy Kids Act was implemented and was without significant change at 43% in 2014–2015 school year (baseline prior to the bundle), but after implementation of the Norwood Bundle, OW/OB decreased to 37% during the 2016-17 school year (*p* = 0.029, value significant). In CPS students, over the same time period, 2014–15 to 2016–17, BMI showed a small but significant rise from 34% to 36% (*p* = 0.013, value significant). Of note, there was little change in the obesity prevalence (22% to 23%) while overweight prevalence declined from 20% to 15%.

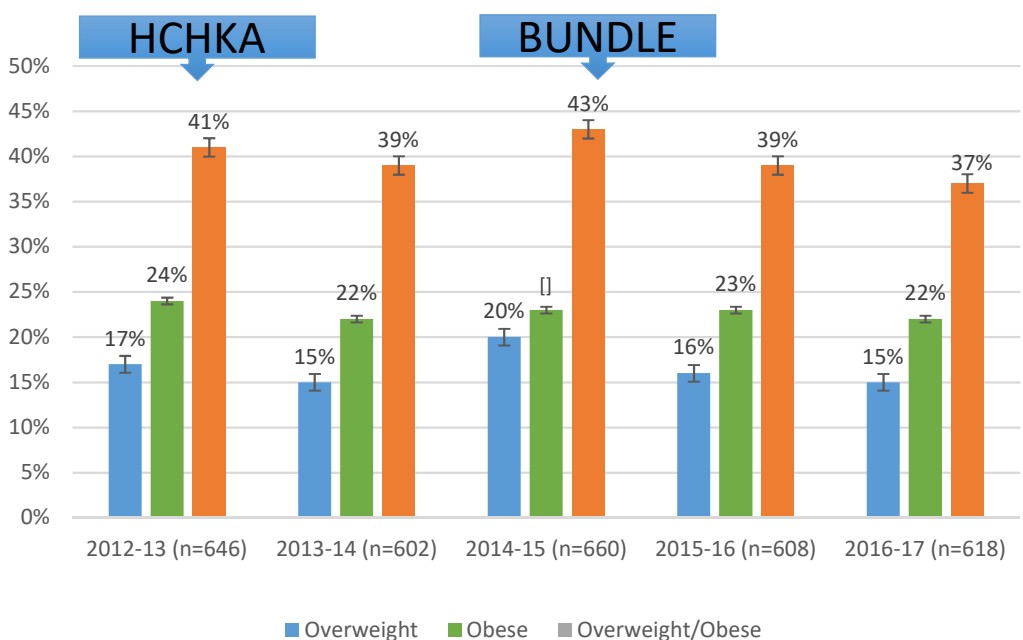

**Figure 1.** BMI status of Norwood City School District students.

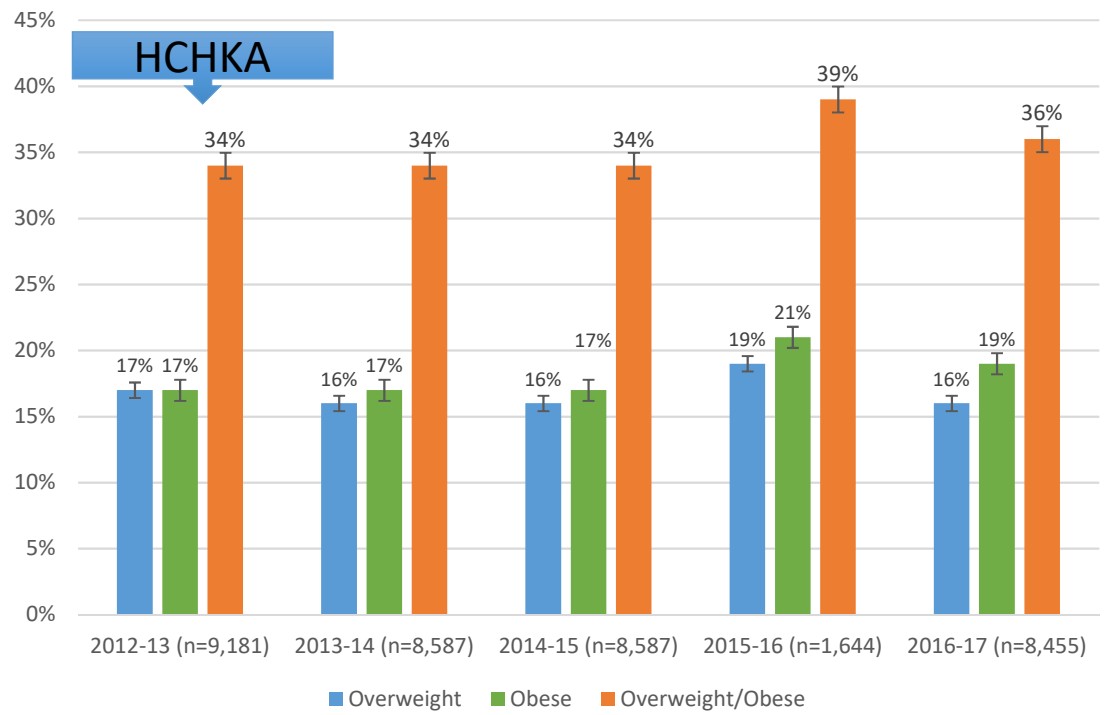

**Figure 2.** BMI status of Cincinnati Publics School students.

## 4. Discussion

With our study, we demonstrate the feasibility of implementing a bundle of interventions directed towards improving the weight status of school children in a small municipality with a population of about 20,000. Like previous community initiatives against obesity, our bundle had multiple components and the primary outcome was an improvement in school-aged children's BMI status. Philadelphia is the

fifth largest city in the US and implemented an extensive city-wide anti-obesity initiative coordinated by the Mayor's office by drawing on the expertise of the City's health department [16]. This multi-faceted effort included the creation of 13 new farmers' markets, increasing bicycle lanes, improving nutritional standards for meals served by city agencies, and a mass media campaign alongside a school health initiative [16,25]. "Get Healthy Philly" resulted in a drop in obesity prevalence from 21.7% to 20.3%. Similarly, "Shape Up Somerville" had components that included school health, home interventions including a newsletter, and community initiatives including walking paths, farmers' market support, media promotion, and health events. Somerville is a community of just over 80,000 [26]. In this study which included two control communities, there was a significant drop in BMI z-score of first- to third-grade students. Our study featured a similar bundle of interventions, but was in a smaller community of about 20,000 and thus with more limited infrastructure and resources. Like the two larger communities we describe, short-term lowering of BMI status of school students is associated with implementing a community bundle directed against obesity. Overall, the reduction of the OW/OB prevalence in Norwood of 6% compares very well to the reduction in prevalence in other communities with bundle interventions, Cambridge 2.4% and Sommerville 0.6% [13,15]. The reduction of obesity prevalence in Norwood of 1% is similar to the 1.4% seen during Philadelphia's bundle intervention [16].

Not surprisingly, as this was a pragmatic community intervention, there are many limitations to the study and what conclusions may be drawn. There was no true control group and there were several background initiatives targeting obesity that touched Norwood including "Let's Move" (nationally), "Healthy Choices for Healthy Children" (statewide), and "We Thrive" (county-wide). We did, however, obtain school BMI data from the larger surrounding municipality's school- children who experienced the same background interventions and showed no improvement in BMI status over the same time frame. For practical reasons, the bundle components were all implemented in a similar time frame and thus we cannot describe the individual effects of each component on the weight status of the Norwood children. Also, like other successful reports from other communities, we do not have long term data to determine if these results are lasting and whether such an effort is sustainable. Additionally, there are inherent inaccuracies in the measure of height and weight in the school setting although there were staff training and a protocol for those doing the measurements. Finally, not all eligible students had their BMI measured each year as parents could opt out and students may have been absent of days BMI was obtained. This is most notable in that CPS had only 1644 students measured in the 2015–16 school year while every other academic year about 8000 were measured which reflects limited resources in getting the measurements that year. Most years, however, participation was good with an estimate of greater than 80% of eligible students getting measured in both school districts.

Anecdotally, there seem to be several lessons learned. When doing a community-wide intervention, there is generally "one shot" and drawing on the success of interventions pioneered by other communities is helpful. Commitment and "buy-in" by the community is essential and ideally, the community should be leading the effort with experts giving advice on what content worked in other communities. With a multipronged approach, it is not possible to determine which components have the most effect. Some components, such as the school-based weight management program, did not directly impact many children. It did, however, establish a very close relationship between the academic medical center, the school district, and the community and fostered the cooperation needed for the design and implementation of the "Norwood Bundle."

## 5. Conclusions

A childhood overweight/obese prevention bundle can be implemented in a small city and is associated with a favorable change in BMI.

**Author Contributions:** A.B. and R.M.S. were involved in the design, implementation, and analysis of the study. They both were involved in writing and revising the manuscript. M.E.H. was involved with study design, implementation and analysis. She was also involved in revising the manuscript. R.K., K.S. and D.R. were involved in study design and implementation and manuscript revision.

**Funding:** This study was funded in part by an Anonymous Family Foundation Grant and a grant from Interact or Health Foundation.

**Acknowledgments:** We gratefully acknowledge the anonymous gift from a family foundation and a grant from Interact for Health for funding elements of the bundle. We also wish to thank Marilyn Crumpton, the Cincinnati Health Department Nurses and all the volunteers involved in measuring the children's BMI.

**Conflicts of Interest:** Siegel has received research funding from Vivus Inc. in the past. The other authors have no potential conflicts of interest to disclose.

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
