# Peer review of "A Community Bundle to Lower School-Aged Obesity Rates in a Small Midwestern City"

_reports, doi:10.3390/reports2030020_

Round 1

Reviewer 1 Report

The authors investigated the effect of multi-component childhood obesity prevention and treatment bundle of interventions in a small mid-western city. However, there are some problems had to be detail clarified 

1. We suggested that the tables were needed to be re-designed again. In the tables, the CI is necessary.

2. Extensive editing of English language is necessary. 

3. There was no formal hypothesis enunciated prior to the study. 

4. This study seems like a scientific report. Can author provide more detail information about the effect of bundle interventions, including body weight or BMI progression.

5. It is necessary to report also all the new studies of the literature on this topic which are now missing. 

6. We suggested authors to compare other bundle intervention or prevention in previous reported articles.

Author Response

Dear Reviewer 1,

Thank you for your careful review and excellent suggestions.   We address all the comments point by point below.

The authors investigated the effect of multi-component childhood obesity prevention and treatment bundle of interventions in a small mid-western city. However, there are some problems had to be detail clarified 

1. We suggested that the tables were needed to be re-designed again. In the tables, the CI is necessary.

Thank you for this excellent suggestion.  Error bars have been added to the bar graphs

Extensive editing of English language is necessary. 

Thank you for noting this.  The text has been carefully reviewed and corrected for typos, tenses and correct style. 

There was no formal hypothesis enunciated prior to the study. 

Thank you for noting this oversight. The hypothesis is now stated in the Abstract lines 20-21 and in the Introduction, lines 47-48 with:

“We hypothesized that implementation of a multi-component bundle would lower the prevalence of OW/OB in Norwood school children.”  

This study seems like a scientific report. Can the author provide more detail information about the effect of bundle interventions, including body weight or BMI progression?

Unfortunately, we were only supplied with aggregate BMI data thus are limited to the results described in the figures.  Also, because all the bundle components were implemented relatively close together we cannot describe the individual effects of each component.   This is now noted as a limitation in the discussion section, lines 175-177 with:

“For practical reasons, the bundle components were all implemented in a similar time frame and thus we cannot describe the individual effects of each component on the weight status of the Norwood children.”

It is necessary to report also all the new studies of the literature on this topic which are now missing. 

Thank you for this suggestion.   The introduction has been expanded with  lines 34-39 and references below:

“The causes of childhood obesity are complex and risk factors include socioeconomic status, environmental factors, inter-uterine exposures, genetic predisposition,  perinatal-antibiotic exposure and  stress. (3-5). Unfortunately, interventions that focus on individual behavior changes have had limited impact and this has led to greater interest in wider, community-based interventions that more definitely address environmental factors. (6,7)”

References:

Liao XP, Yu Y, Marc I, Dubois L, Abdelouahab N, Bouchard L, Wu YT, Ouyang F, Huang HF, Fraser WD. Prenatal determinants of childhood obesity: a review of risk factors. Canadian journal of physiology and pharmacology. 2019 Jan 19;97(3):147-54.

Grassi T, De Donno A, Bagordo F, Serio F, Piscitelli P, Ceretti E, Zani C, Viola G, Villarini M, Moretti M, Levorato S. Socio-Economic and environmental factors associated with overweight and obesity in children aged 6–8 years living in five Italian cities (the MAPEC_LIFE cohort). International journal of environmental research and public health. 2016 Oct 11;13(10):1002.

Weihrauch-Blüher, Susann, and Susanna Wiegand. "Risk factors and implications of childhood obesity." Current obesity reports 7.4 (2018): 254-259.

Ananthapavan J, Nguyen PK, Bowe SJ, Sacks G, Herrera AM, Swinburn B, Brown V, Sweeney R, Lal A, Strugnell C, Moodie M. Cost-effectiveness of community-based childhood obesity prevention interventions in Australia. International Journal of Obesity. 2019 May;43(5):1102.

Brown T, Moore TH, Hooper L, Gao Y, Zayegh A, Ijaz S, Elwenspoek M, Foxen SC, Magee L, O'Malley C, Waters E. Interventions for preventing obesity in children. Cochrane Database of Systematic Reviews. 2019(7).

We suggested authors to compare other bundle intervention or prevention in previous reported articles.

Thanks for this excellent suggestion.  Our intervention bundle’s change in overweight/obesity is now compared to the interventions in 3 other communities in lines 163-166 with:

“Overall, the reduction of the OW/OB  prevalence in Norwood of 6% compares very well to the reduction in prevalence in other communities with bundle interventions, Cambridge 2.4% and Sommerville 0.6%. (13,15)   The reduction of obesity prevalence in Norwood of 1% is similar to the 1.4% seen during Philadelphia’s bundle intervention. (16)”

Reviewer 2 Report

In the manuscript “A Community Bundle to Lower School-Aged Obesity Rates in a Small Mid-Western City” the authors aimed to describe the effects of a community bundle on childhood overweight/obesity rates in a small city where the resources are limited in respect of larger cities. The results showed that there was a positive impact with a favorable change in children’s BMI. The work is good but it would be more interesting if it was better written. As reported in the limitations section, no long term data are available to determine if the results are lasting and whether such an effort is sustainable. However, this study may represent a good example to realize the goal of a healthy society, starting from children in small cities.

Therefore, I believe that the work, with some minor revision (indicated below), may be published. 

Main notes:

1. In general, the form of the text must be improved (punctuation, capitalization, tenses, etc.) and there are some spelling errors. Acronyms are often written without spelling out the full names of institutions or programs, so it would be advisable to insert the full name followed by the acronym and then using it only. Moreover, some acronyms are wrong.

2. In the abstract the value of pis not the same reported in the text. 

3. The introduction does not provide sufficient background. It should be broadened by better defining the etiology, risks and factors involved in the onset of childhood obesity (i.e. Grassi et al. International Journal of Environmental Research and Public Health, 2016; Weihrauch-Blüher and Wiegand, Current Obesity Reports, 2018; Liao et al. Canadian Journal of Physiology and Pharmacology, 2019).

4. Methods should be improved describing how the anthropometric measurements of children were taken and how BMI was calculated. In addition:

· Line 62: “A level three obesity clinic”. I think that the different levels of obesity should be explained in order to let the readers better understand. 

· Lines 64-65: “The clinic met monthly accommodating up to 4 students and their guardians a clinic.” This sentence should be better formulated. 

5. In the results section, it should be underlined that only the rate of overweight children was decreased but not the obesity one. 

Author Response

Dear Reviewer 2,

Thank you for your careful review and all the helpful suggestions. The comments and the corrections are outlined below:

In general, the form of the text must be improved (punctuation, capitalization, tenses, etc.) and there are some spelling errors. Acronyms are often written without spelling out the full names of institutions or programs, so it would be advisable to insert the full name followed by the acronym and then using it only. Moreover, some acronyms are wrong.

The text has been carefully checked for all these issues and corrected.  Thank you for pointing out these issues.

In the abstract the value of p is not the same reported in the text. 

This error has been corrected. 

The introduction does not provide sufficient background. It should be broadened by better defining the etiology, risks and factors involved in the onset of childhood obesity (i.e. Grassi et al. International Journal of Environmental Research and Public Health, 2016; Weihrauch-Blüher and Wiegand, Current Obesity Reports, 2018; Liao et al. Canadian Journal of Physiology and Pharmacology, 2019).

Thank you for this suggestion.  These references along with two others have been added to the background in the Introduction in lines 34-39

“The causes of childhood obesity are complex and risk factors include socioeconomic status, environmental factors, inter-uterine exposures, genetic predisposition,  perinatal-antibiotic exposure and  stress. (3-5). Unfortunately, interventions that focus on individual behavior changes have had limited impact and this has led to greater interest in wider, community-based interventions that more definitely address environmental factors. (6,7)”

References:

Liao XP, Yu Y, Marc I, Dubois L, Abdelouahab N, Bouchard L, Wu YT, Ouyang F, Huang HF, Fraser WD. Prenatal determinants of childhood obesity: a review of risk factors. Canadian journal of physiology and pharmacology. 2019 Jan 19;97(3):147-54.

Grassi T, De Donno A, Bagordo F, Serio F, Piscitelli P, Ceretti E, Zani C, Viola G, Villarini M, Moretti M, Levorato S. Socio-Economic and environmental factors associated with overweight and obesity in children aged 6–8 years living in five Italian cities (the MAPEC_LIFE cohort). International journal of environmental research and public health. 2016 Oct 11;13(10):1002.

Weihrauch-Blüher, Susann, and Susanna Wiegand. "Risk factors and implications of childhood obesity." Current obesity reports 7.4 (2018): 254-259.

Ananthapavan J, Nguyen PK, Bowe SJ, Sacks G, Herrera AM, Swinburn B, Brown V, Sweeney R, Lal A, Strugnell C, Moodie M. Cost-effectiveness of community-based childhood obesity prevention interventions in Australia. International Journal of Obesity. 2019 May;43(5):1102.

Brown T, Moore TH, Hooper L, Gao Y, Zayegh A, Ijaz S, Elwenspoek M, Foxen SC, Magee L, O'Malley C, Waters E. Interventions for preventing obesity in children. Cochrane Database of Systematic Reviews. 2019(7).

Methods should be improved describing how the anthropometric measurements of children were taken and how BMI was calculated. In addition:

This is now described in the Methods lines 114-120 with:

“BMI data for students in grades K, 1, 3,5, and 9 were made available for NCSD and the surrounding school district, CPS, through the “Healthy Choices for Healthy Kids Act.  Height and weight were measured with the children in light clothing and no shoes using a portable scale and stadiometer (SECA, Hanover, New Hampshire).  Overweight was defined by the Center for Disease Control and Prevention (CDC )recommendation of the 85th percentile of BMI for age and sex to < the 95th percentile and obesity as > the 95th percentile; the children’s weight status was determined using the excel spreadsheet offered by the CDC. (16)”

Line 62: “A level three obesity clinic”. I think that the different levels of obesity should be explained in order to let the readers better understand. 

This is now clarified in lines 68-72 with:

The School CBHN Clinic-The Center for Better Health and Nutrition (CBHN) is the pediatric weight management at Cincinnati Children’s Hospital and was invited by Norwood City School District (NCSD) to bring their program to the centrally located high school building.  NSCD has 3 elementary schools, a middle school, and a high school.  A stage three structured pediatric weight management clinic was established as described by the American Academy of Pediatrics with a CBHN multi-disciplinary team of nurse, pediatrician, dietitian and exercise physiologist. (14)”

Lines 64-65: “The clinic met monthly accommodating up to 4 students and their guardians a clinic.” This sentence should be better formulated. 

Thank you for noting the confusion.  Lines 74 to 76 now reads:

“The clinic was held monthly and up to 4 students and their guardians were cared for at each clinic session.”

In the results section, it should be underlined that only the rate of overweight children was decreased but not the obesity one. 

Thank you for this suggestion.  This is now noted with lines 138-139 with:

“Of note, there was little change in the obesity prevalence (22 to 23%) while overweight prevalence declined from 20 to 15%.”

Round 2

Reviewer 1 Report

We thanks authors answer the questions point to point and provide the detail information.